# Isolation, Structural Characterization, and Biological Activity of the Two Acidic Polysaccharides from the Fruits of the *Elaeagnus angustifolia* Linnaeus

**DOI:** 10.3390/molecules27196415

**Published:** 2022-09-28

**Authors:** Haibaier Huojiaaihemaiti, Paiheerding Mutaillifu, Adil Omer, Rehebati Nuerxiati, Xiaomei Duan, Xuelei Xin, Abulimiti Yili

**Affiliations:** 1State Key Laboratory Basis of Xinjiang Indigenous Medicinal Plants Resource Utilization, and the Key Laboratory of Chemistry of Plant Resources in Arid Regions, Xinjiang Technical Institute of Physics and Chemistry, Chinese Academy of Sciences, South Beijing Road 40-1, Urumqi 830011, China; 2University of Chinese Academy of Sciences, Beijing 100039, China

**Keywords:** oleaster, acidic polysaccharides, structural elucidation, antioxidant activity and immuno-regulatory

## Abstract

*Elaeagnus angustifolia* Linnaeus is a medicinal plant and its fruit has pharmacological activity such as antiinflammatory, antiedema, antinociceptive, and muscle relaxant functions, etc. Two acidic homogeneous polysaccharides (EAP-H-a1 and EAP-H-a2) were isolated from the fruits of *Elaeagnus angustifolia* L. through DEAE-52 and Sephadex G-75 column chromatography, and the physicochemical, structural properties, and biological activities of the polysaccharides were investigated. Both EAP-H-a1 and EAP-H-a2 were composed of Rha, Ara, Xyl, Glc, and Gal with the molar ratios of 13.7:20.5:23.3:8.8:33.4 and 24.8:19.7:8.2:8.4:38.6, respectively, and with the molecular weights of 705.796 kDa and 439.852 kDa, respectively. The results obtained from Fourier transform infrared spectroscopy (FTIR) confirmed the polysaccharide nature of the isolated substances. Congo red assay confirmed the existence of a triple-helix structure. Scanning electron microscopy (SEM) and X-ray diffraction (XRD) analysis revealed that EAP-H-a1 and EAP-H-a2 had irregular fibrous, filament-like surfaces; and both had crystalline and amorphous structures. Bioactivity analysis showed that the crude polysaccharide, EAP-H-a1, and EAP-H-a2 had clear DPPH and ABTS free radical scavenging activity, and could promote the secretion of NO and the phagocytic activities of RAW 264.7 and THP cells, which showed clear antioxidant and immuno-regulatory activity. These results indicated that *Elaeagnus angustifolia* L fruit acidic polysaccharides may have potential value in the pharmaceutical and functional food industries.

## 1. Introduction

*Elaeagnus* is a genus that belongs to the *Elaeagnaceae* family that has 55 species in China [1] and three types in the Xinjiang Autonomous Region, China, namely, *Elaeagnus angustifolia* Linnaeus, *Elaeagnus Oxycarpa* Schlecht, and *Elaeagnus mooceroftii* Wall [2]. *Elaeagnus angustifolia* is called Jigde in the Uyghur language and is collected in the drug standard of the Chinese Ministry of Health. In Uyghur traditional medicine, the fruits with peels without seeds of E. *angustifolia* were used for the treatment of diarrhea, premature ejaculation, gastric ulcers, and leucorrhea. The jam made from flowers was used as a functional food to prevent female erectile dysfunction and antiinflammation [3]. The flower honey from *E. angustifolia* L. was used to expel coldness. The gum polysaccharides from the tree stem were used as an ingredient in hair tonic and alleviated the impairment of the barrier function in dry skin [3,4,5].

Phytochemical reports have shown that the fruits have multiple chemical constituents, including proteins, amino acids, peptides, polysaccharides, monosaccharides, flavonoids, and inorganic elements with numerous biological activities such as antiinflammatory, antiedema, antinociceptive, and muscle relaxant functions, etc. [4,6]. Polysaccharides, as one of the main bioactive components of this medicinal plant, have shown numerous biological activities, such as antioxidant, antibacterial, antitumor, and immune-regulating functions, and obtained a variety of applications in the pharmaceutical, cosmetic, and functional food industry [7,8,9,10,11]. To the best of our knowledge, studies concerned with *E. angustifolia* L. fruit polysaccharides have concentrated on neutral polysaccharide [7,8,9,10,11]. However, there is no research about the systematic isolation, structural characterization, and biological activities of acidic polysaccharide composition from the fruits of *E. angustifolia* L. [7]. Therefore, the purpose of the present study was to isolate, structurally elucidate, and evaluate the biological activity of this medicinal plant.

In this paper, polysaccharides were isolated through sequential water extraction, DEAE-52, and Sephadex G-75 column chromatography, and the physicochemical, as well as structural properties, were evaluated under the combination of extensive analysis such as HPGPC, GC, GC–MS, NMR, TG, DTG, DSC, XRD, and Congo red staining. Meanwhile, antioxidant and immuno-regulatory activities of the obtained polysaccharides were evaluated. This study could provide the basis for the further utilization of *E. angustifolia* L.

## 2. Results and Discussion

### 2.1. Chemical Composition of the Polysaccharides

One neutral and two acidic polysaccharide fractions were isolated from the crude EAP-H through DEAE-52 anion exchange column chromatography, namely, EAP-N, EAP-H-a1, and EAP-H-a2 (Figure 1). Then, all three fractions were further purified by Sephadex G-75 gel permeation column chromatography, and a single and symmetric peak at the elution profile of each fraction was obtained (Figure 1). However, the results of homogeneity analysis through HPLC showed there was a single peak for EAP-H-a1 and EAP-H-a2 (Figure 2), showing that both EAP-H-a1 and EAP-H-a2 were close to the homogenous polysaccharide, and EAP-N needed further purification. Then, the structural characteristics of EAP-H-a1 and EAP-H-a2 were analyzed.

The intensive positive value of specific optical rotation (Table 1) suggests that α-configuration predominates over β-configuration in EAP-H-a1 and EAP-H-a2 [12]. The neutral sugar, uronic acid, protein content, Mw, and monosaccharide composition of EAP-H-a1 and EAP-H-a2 are summarized in Table 1, and there was no significant difference between the neutral sugar and uronic acid content of EAP-H-a1 and EAP-H-a2 (*p* > 0.05). The high proportion of uronic acid also manifested in that both EAP-H-a1 and EAP-H-a2 were acidic fractions. The Mw values of EAP-H-a1 and EAP-H-a 2 were determined to be 705.796 and 439.852 kDa, respectively, by HPGPC analysis, and the Mw of EAP-H-a1 was significantly higher than EAP-H-a2 (*p* < 0.05), which showed the structure of EAP-H-a1 might be more complicated than that of EAP-H-a2. Both of the two fractions were composed of Rha, Ara, Xyl, Glc, and Gal (Figure 2). Gal was the predominant monosaccharide for both EAP-H-a1 and EAP-H-a2, followed by Xyl and Ara for EAP-H-a1, and Rha and Ara for EAP-H-a2. Liu et al. [13] isolated a homogenous polysaccharide EAP-1a from *Elaeagnus angustifolia* L. with 61.42 kDa through hot water extraction, and it was mainly composed of Ara, Xyl, Glc, and Gal with a molar ratio of 0.74:0.51:1:5.19, respectively. PEA-1 and PEA-2, isolated from *Elaeagnus angustifolia* L. through water extraction, were reported to have Mw values of 9113 and 5020 Da, respectively, and PEA-1 was mainly composed of Rha, Xyl, Man, Glc, and Gal, while Rha, Man, Glc, and Gal composed PEA-2 [14].

### 2.2. Structural Characterization of the Polysaccharides

#### 2.2.1. UV and FTIR Analysis

There is no optical absorption of the sample in the UV spectrum at 260 or 280 nm, which is in accordance with the protein content analysis in the BCA assay. FTIR spectroscopic analysis was recorded from 4000 to 400 cm^−1^ wavelengths (Figure 3). There is a broadly-stretched intense peak at approximately 3420 cm^−1^ indicating O-H stretching vibrations, and the absorption peak at about 2926 cm^−1^ represents the existence of an asymmetric bending vibration caused by the C-H bond [15]. There is a strong absorption peak around 1726 cm^−1^ attributed to the existence of the carboxyl group, which suggests that both EAP-H-a1 and EAP-H-a2 are acidic polysaccharides [16]. The peak around 1680 cm^−1^ is the signal of associated water [17]. A strong band around 1200–1000 cm^−1^ is caused by the stretching vibrations of C-O side groups and the C-O-C glycosidic band vibrations, which indicate the possible presence of the pyranose ring [18]. The weak absorption at 830 cm^−1^ and 925 cm^−1^ indicates the existence of both α and β configurations [19,20].

#### 2.2.2. Conformational Analysis

Generally, triple-helix polysaccharides combined with Congo red lead to the redshift of the λ_max_ of the Congo red–polysaccharide complex [21]. Therefore, Congo red assay was used to determine the triple-helix structures of the polysaccharides. No redshift for the λ_max_ of the complex with different concentrations (0–0.50 mol/L) of NaOH was observed in EAP-H-a1 and EAP-H-a2 (Figure 4), indicating neither contained a triple-helical structure.

#### 2.2.3. Morphological Properties

Scanning electron microscopy (SEM) was considered to be the most effective analytical technique for structural morphology analysis including the size, shape, and porosity of macromolecules. The SEM images exhibit that both of the obtained polysaccharides have different structures (Figure 5). EAP-H-a1 has a fibrous surface-attached spherical microsphere-like structure, while EAP-H-a2 shows a ribbon with a microporous structure and a fibrous surface. Meanwhile, the differences in the surface morphology of the polysaccharides might have been caused by the different methods used to prepare them, including extraction and purification methods, which is in agreement with the reported literature [22].

#### 2.2.4. X-ray Diffraction Analysis

The XRD profile exhibited the characteristic diffraction curve of each EAP purified polysaccharide. The XRD profile gave the characteristic diffraction curve. As shown in Figure 6, the major reflection suggests the presence of both crystalline and amorphous structures of polysaccharides. EAP-H-a-1 has diffraction peaks at approximately 11°, 12°, 21°, 22°, 25°, 26°, and 31°. EAP-H-a1 presents more crystalline than EAP-H-a2, which has diffraction peaks at approximately 12.5°, 18°, and 20.4°, and the comparative results demonstrate the influence of the extraction process on the structure of the polysaccharides.

#### 2.2.5. Thermal Properties

The results of the thermal properties of the two polysaccharides are shown in Figure 7. According to the TG curve, polysaccharide fractions had a slight weight loss (10%) between 80 °C and 120 °C, and a major weight loss occurred between 210 °C and 500 °C. Residual mass was 30% when the temperature rose to 500 °C. Two peak weight loss temperatures were seen at 98 °C and 240 °C determined by a DTG derivative weight loss curve. According to the DSC curve, at the beginning the polysaccharide fraction absorbs heat, then it begins to release heat gradually including an exothermic steep at the range of 250–450 °C. There was one major endothermic peak around 250 °C during decomposition between EAP-H-a1 and EAP-H-a2. There was no difference related to thermal properties.

#### 2.2.6. NMR Results

Six obvious chemical shifts of the anomeric proton were found in the anomeric region at 5.17, 5.11, 5.04, 5.00, 4.99, and 4.98 ppm for EAP-H-a2, which were higher or lower than 5.0 ppm, indicating the existence of both α- and β-configurations [23], which was further complemented by the results of the FTIR spectrum. The strong chemical shift at 4.80 ppm can be attributed to the solvent D_2_O. The chemical shift appearing around 1.26 ppm indicates the existence of Rha residue (Figure 8).

The ^13^C NMR spectrum shows characteristic signals at 174.87 ppm indicating the presence of uronic acid [24]. The presence of signals at 90–110 ppm indicates that all of the sugar residues were in the form of both α- and β-configurations. There are also six obvious chemical shifts of anomeric carbon found at 100.66, 102.58, 103.26, 104.25, 107.46, 107.19, and 108.08 ppm. There were also observed anomeric carbons with chemical shifts higher and lower than 103 ppm, indicating the existence of α and β configurations of EAP-H-a2; this was in accordance with the results of ^1^H NMR.

### 2.3. Biological Activity

#### 2.3.1. Antioxidant Activity

The DPPH and ABTS free radical scavenging activity of EAP-H-a1 and EAP-H-a2 are shown in Figure 9. The DPPH radical scavenging activity of Vc was significantly higher than that of polysaccharides; all polysaccharides had dose-dependent DPPH scavenging activity (Figure 9a). EAP-H-a1 had higher DPPH free radical scavenging activity than EAP-H-a2. The results of ABTS free radical scavenging are shown in Figure 9b indicating that crude polysaccharide exhibits significantly higher scavenging activity of ABTS free radicals than that of Vc. All polysaccharides exhibited a dose-dependent pattern at concentrations ranging from 0.2 to 1 mg/mL. The DPPH radical scavenging activity of EAP-H-a1 and EAP-H-a2 was weaker than that of PEA-1 and PEA-2, and the activity of crude EAP was higher than that of the polysaccharide obtained from *Elaeagnus angustifolia* by Chen et al. [9,14]. The differences between the antioxidant activity of the polysaccharides may be caused by the different extraction methods, the monosaccharide composition, Mw, and the configuration, etc., and this result is in accordance with the previous reports [25,26].

#### 2.3.2. MTT Cell Viability

The macrophage is one of the critical parts of the natural immune defense system of hosts, and has various immune regulatory functions [27]. Furthermore, immuno-regulatory activity is one of the main biological activities of polysaccharides; numerous plant-originated polysaccharides have demonstrated immuno-modulatory activity to maintain the host’s health by regulating the immune response [28,29,30]. 

The cytotoxic activities of EAP-H-C and its two purified fractions were evaluated on RAW 264.7 and THP-1 cells. As is shown in Figure 10a, the cell viability of EAP-H-C and EAP-H-a2 against RAW 264.7 decreased between the sample concentrations of 200–800 µg/mL in a dose-dependent manner, and the cytotoxicity of EAP-H-C was significantly lower than EAP-H-a1 and EAP-H-a2. However, the cell viability of polysaccharides against THP-1 was quite different from RAW 264.7. It showed irregular changes with the increase in concentration (Figure 10b). The results showed EAP-H-a1 had no obvious cytotoxicity against RAW 264.7 and EAP-H-C against THP-1 cells.

#### 2.3.3. Effects of Polysaccharides on Macrophage Activation

Macrophage activation is regarded as one of the novel immunotherapeutic approaches for the treatment of many diseases. Once macrophages are activated, they can release a significant amount of NO, which contributes to the killing of tumor cells and pathogenic microorganisms and acts as an intracellular messenger molecule to mediate a variety of biological functions, and known as a vital active substance related to the immunomodulating response to macrophage cells [31].

The effects of polysaccharides on NO production of RAW 264.7 cells are shown in Figure 10c. The NO secretion of all polysaccharides was significantly decreased compared with LPS, indicating EAP-H-C and its two purified fractions had potential immuno-regulating activity. When the concentration was in the range of 25–100 µg/mL, the production of NO increased significantly after being cultivated with three polysaccharides for 24 h in a dose-dependent manner. The following increase in concentration caused irregular changes in the NO production. The results confirmed that EAP-H-C had the function of the activation of macrophages and dose-dependently increased NO release from RAW 264.7 cells. Polysaccharides do not affect NO production when it is concerned with THP-1.

#### 2.3.4. Effect of EAP on Phagocytosis of Macrophages

Phagocytic capacity is a primary indicator of the effects of macrophage activation. In this work, the phagocytic activity of RAW 264.7 and THP-1 cells treated with EAP-H-C, EAP-H-a1, and EAP-H-a2 were evaluated by the neutral red test. Phagocytic activities of the three polysaccharides increased within the test concentrations. Compared with the blank control group, the phagocytic activities of RAW 264.7 treated with EAP-H for 24 h were enhanced significantly (*p* < 0.05) in a dose-dependent manner (Figure 10d). When the EAP-H-a1 concentration was 50 µg/mL, there was no significant difference (*p* > 0.05) between the sample group and the positive control. EAP-H-a2 enhanced phagocytic capacity significantly in a dose-dependent manner. Compared with the blank control group, the phagocytic activities of THP-1 treated with EAP-H for 24 h were enhanced significantly (*p* < 0.05). When the EAP-H concentration was 50 µg/mL, there was no significant difference (*p* > 0.05) between the sample group and the LPS (Figure 10e).

## 3. Materials and Methods

### 3.1. Materials and Reagents

Dried *Elaeagnus angustifolia* L. fruits were collected in Hotan, Xinjiang Uyghur Autonomous Region, China, and dried at room temperature. The fruit of *Elaeagnus angustifolia* L. was identified by Lu Chunfang, associate professor of Xinjiang Institute of physical and chemical technology, Chinese Academy of Sciences. Voucher specimens (WY02680) have been deposited in Xinjiang Institute of Physics and Chemistry, Chinese Academy of Sciences. The pulp was separated by soaking the fruit in ethanol and the seeds were separated manually. 

Dialysis bags were purchased from BioSharp (Hefei, China). Dextran standards and DPPH were purchased from Sigma-Aldrich (Saint Louisi, MO, USA). DEAE-52, Sephadex G-75, and ABTS were obtained from Solarbio (Beijing, China). 3-(4, 5-dimethylthiazol-2-yl)-2, 5- diphenyltetrazolium bromide (MTT) was purchased from Sigma Co. (St. Louis, MO, USA). RAW 264.7 and THP-1 cells were acquired from the Cell Bank of Shanghai Institute of Cell Biology (Shanghai, China). Griess Reagent was purchased from Bi Yun Tan Co., Ltd. (Shanghai, China). All other reagents were analytical reagents.

### 3.2. Pretreatment and Sequential Extraction of the Polysaccharides

The small molecules of the dried fruits of *E. Angustifolia* L. were removed through soaking in 95% ethanol at room temperature for 24 h. The obtained powder (1000 g) was extracted by sequential extraction with water (room temperature 25 °C), hot water (85 °C), acid, and alkaline, respectively. Firstly, the extraction was carried out with water at room temperature with the material–liquid ratio of 1:30 g/mL for 2 h and obtained EAP-C. Then, the residue was submitted to the next extraction by hot water at 90 °C with the solid–liquid ratio of 1:30 g/mL for 2 h and obtained EAP-H. The third extraction was carried out with aqueous oxalic acid (0.5%) and ammonium oxalate (0.5%) at room temperature (pH 2.50, 1:30 g/mL, 2 h) for EAP-P, and the last one was extracted with 1 M NaOH at 8 °C (1:30 g/mL, 2 h), and the extracts neutralized with 50% acetic acid, and dialyzed 48 h, which obtained EAP-He. The extracts were centrifuged (10,000 rpm, 10 min) and the supernatant was concentrated on a rotary evaporator. All the prepared extracts were precipitated with 4-fold excess ethanol. According to the yield and polysaccharide content, the EAP-H was further purified by column chromatography [32].

### 3.3. Purification of the EAP-H

The EAP-H was dissolved in distilled water and dialyzed (cut-off 3500 Da) against water for 2 days, then deproteinized with the Sevag method, and the resulting aqueous phase was lyophilized. A total of 100 mg of deproteinized EAP-H was dissolved in 10 mL distilled water and applied to the DEAE-52 column and eluted with distilled water and different concentrations of NaCl solution (0.2 M, 0.4 M, 0.6 M, and 0.8 M) with the flow rate of 0.6 mL/min, then collected by semi-automatic collector, and monitored with the anthrone–sulfuric method. The obtained fractions were collected according to the elution curve; among them, the ones eluted with NaCl solution were dialyzed for 48 h and lyophilized. Then, the fractions were further fractionated by using a Sephadex G-75 column (1.6 × 70 cm) and eluted with ultra-pure water with a flow rate of 0.1 mL/min to obtain homogenous polysaccharide.

### 3.4. Structural Characterization

#### 3.4.1. The Specific Optical Rotations

The specific optical rotations ([α]20 D) of purified polysaccharides were measured at sodium D line 589 nm using a AUTOPOL VI automatic polarimeter (RUDOLPH RESEARCH ANALYTICAL) at 20 °C. The purified polysaccharides were dissolved in distilled water (1.0 mg/mL) and put in 1 dm liquid-holding tubes.

#### 3.4.2. Monosaccharide Composition Analysis

Monosaccharide composition of polysaccharides was determined by GC–MS (7890A-5975C, Agilent Technologies, Palo Alto, Santa Clara, CA, USA) chromatography according to our previous reports [33].

#### 3.4.3. Molecular Weight Determination and Homogeneity Analysis

The molecular weight (Mw) of the purified polysaccharides was determined using high-performance liquid chromatography equipped with a TSK-G3000 PW_XL_ column (TOSHO, Japan) and differential refractive index detector (RID-10A, Shimadzu, Japan) according to our previous reports [34].

#### 3.4.4. UV and FTIR Spectroscopy Analysis

The UV absorption pattern of the purified polysaccharides between 200 and 400 nm was detected using a 2550 UV spectrophotometer (Shimadzu, Japan). A NICOLET 6700 Fourier transform infrared spectrophotometer (Thermo, Waltham, MA, USA) was used to observe the functional groups of polysaccharides in the range of 4000–400 cm^−1^.

#### 3.4.5. SEM Analysis

A scanning electron microscope (SUPRA 55VP, ZEISS, Germany) was used to observe the surface structure of polysaccharides with magnification at 500–1000× under a high vacuum at a voltage of 20.0 kV.

#### 3.4.6. XRD Analysis

A D8 Advance X-ray powder diffractometer (Bruker, Germany) was used to detect the crystal structure of polysaccharides, 2θ ranging from 5 to 60° at room temperature [35].

#### 3.4.7. Analysis of Thermogravimetric and Differential Scanning Calorimetry

Thermal analysis (TGA), differential thermal gravity (DTG), and differential scanning calorimetry (DSC) of the polysaccharides were conducted on a thermogravimetric analyzer (STA449F3, Netzsch, Germany). A 2 mg sample was placed in the sample pan and heated from 30 °C to 600 °C at 10 °C/min under a protective nitrogen atmosphere (20 mL/min), and nitrogen was used at a flow rate of 50 mL/min [36].

#### 3.4.8. Congo Red Analysis

The conformational characteristics of the purified polysaccharides were analyzed by the Congo red analysis. A 2 mg/mL concentration of sample solution was prepared, then 1 mL sample, 1 mL Congo red reagent (0.2 mmol/L), and 1 mol/L NaOH solution of different volumes were mixed; the final concentrations of NaOH were 0.5, 0.4, 0.3, 0.2, 0.1, and 0.05 mol/L, and they were reacted at room temperature for 5 min. The maximum absorption wavelength (λ_Max_) of the reaction solution in the 400–700 nm range was measured by a UV spectrophotometer. Congo red reagent and different concentrations of NaOH solution without sample solution were used as the control group [37].

#### 3.4.9. NMR Analysis

In total, 40 mg of polysaccharide was dissolved in 1 mL of 99.9% deuterium oxide and the sample was placed in NMR tubes. The ^1^H NMR and ^13^C NMR spectra were recorded on an NMR spectrometer (Varian VNMRS 600 spectrometer, Palo Alto, CA, USA) [38].

### 3.5. Antioxidant Activity

The DPPH and ABTS free radicals scavenging activity of the polysaccharides were conducted according to our reported method [38]. DPPH test described briefly: 1 mL of different concentrations of polysaccharides in water (0.25–2.0 mg/mL) was mixed with 1 mL of freshly prepared DPPH in methanol (0.2 mmol/L). The mixture was shaken and stored at 37 °C for 30 min in dark. The absorbance was measured at 517 nm. Vc was used as the positive control. ABTS test described briefly, 7 mmol/L of ABTS solution mixed with 2.45 mmol/L of potassium persulfate and stored in the dark at room temperature for 16 h to prepare ABTS radical. At the moment of use, the ABTS radical solution was diluted with ethanol to an absorbance of 0.70 ± 0.02 at 734 nm. An amount of 0.3 mL of different concentrations of polysaccharides in water (0.1–4.0 mg/mL) was added to 2.7 mL of ABTS radical solution and mixed, then reacted at room temperature for 6 min, absorbance at 734 nm was measured. 

### 3.6. Cytotoxicity Assay

The cytotoxicity assay of purified polysaccharides was carried out according to a previous study [36]. Briefly, Dulbecco’s modified Eagle’s medium (DMEM, Gibco) was mixed with fetal calf serum (10%) and penicillin (100 U/mL)–streptomycin (0.1 mg/mL). The RAW 264.7 and L6 cells were cultured for 24 h in DMEM under a suitable condition (37 °C, 5% CO_2_). After incubation with polysaccharide for 24 h, 20 μL of MTT (5 mg/mL) was added to each well. Lastly, the MTT solution was removed and 200 μL DMSO was added. Finally, absorbance was recorded at 570 nm.

### 3.7. Assay of Macrophage Phagocytosis

The phagocytic ability of RAW 264.7 macrophages was determined using the neutral red uptake method according to previous reports with modifications [31]. Briefly, cells (1 × 10^6^ cells/well) were dispensed in 96-well plates and incubated for 12 h. Then, RAW 264.7 cells were treated with different concentrations of polysaccharides (25, 50, 100, 200, and 400 g/mL), respectively, and continued to be incubated for 24 h. The medium and LPS (2 g/mL) were used as a blank and positive control, respectively. Subsequently, 0.075% neutral red solution (100 μL/well) was added and incubated for 1 h. The medium was discarded and cells were washed three times with PBS. Afterward, cell lysis buffer (1% glacial acetic acid–ethanol = 1:1, 100 μL/well) was added to lyse cells. When cells were incubated at room temperature for 15 h, the optical density was measured at 540 nm on a microplate reader (Spectra Max M5, Thermo Scientific, Waltham, MA, USA), and the phagocytosis index (PI) was calculated.

### 3.8. Antiinflammatory Assay

The antiinflammatory assay of the polysaccharides was evaluated by measuring the secretion of NO in RAW 264.7 cells induced by lipopolysaccharide (LPS), Griess reaction was utilized according to previous reports [36].

## 4. Conclusions

In this study, sequential extractions of polysaccharides from *E**. angustifolia* L. were performed and the hot water extracted from one was further purified by anion exchange and gel permeation chromatography. The valuable structural information about the physicochemical properties of the isolated two acidic polysaccharides was obtained by using different analytical tools and methods. EAP-H-a1 and EAP-H-a2 were homogenous acidic polysaccharides with an average Mw of 705.796 kDa and 439.852 kDa, respectively, which composed of Rha, Xyl, Ara, Man, Glc, and Gal in different molar ratios. Both have certain free radical scavenging activity against DPPH and ABTS, demonstrating their antioxidant function. The obtained acidic polysaccharides could promote the secretion of NO and phagocytic activities of RAW 264.7 and THP-1. The overall study suggested that acidic EAP-H could be used as a potential natural antioxidant and immuno-regulating drug resource and has value for further scientific research in the food and pharmaceutical industries. 

## Figures and Tables

**Figure 1 molecules-27-06415-f001:**
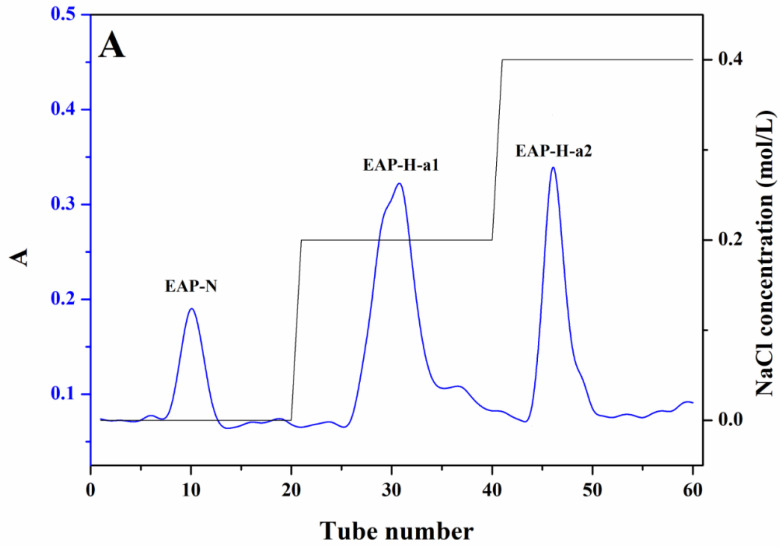
Stepwise elution curves of the polysaccharide fractions using DEAE Sepharose-52 chromatography (**A**); Sephadex G-75 chromatography elution curve of different fractions (**B**,**C**).

**Figure 2 molecules-27-06415-f002:**
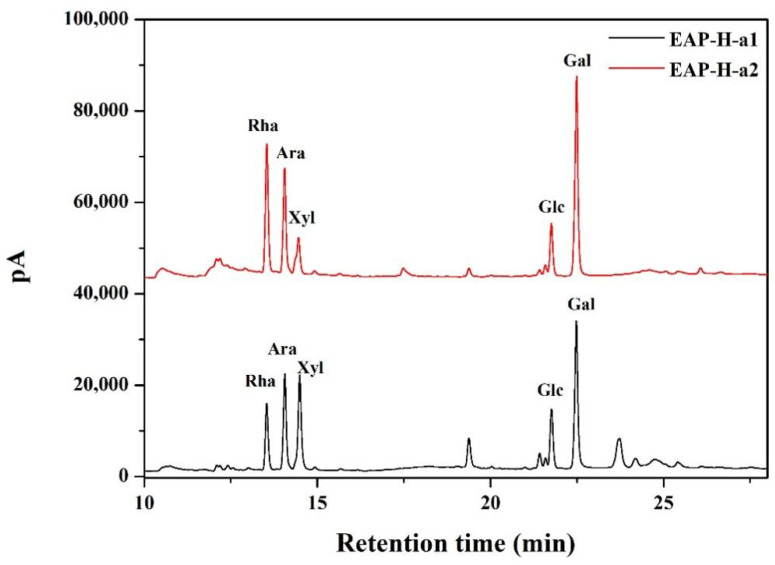
GC spectrum of the polysaccharides.

**Figure 3 molecules-27-06415-f003:**
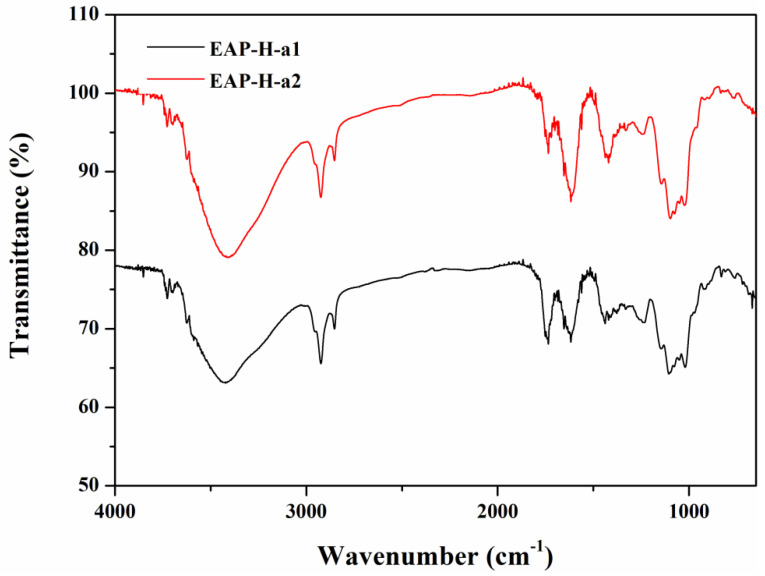
IR spectrum of the polysaccharides.

**Figure 4 molecules-27-06415-f004:**
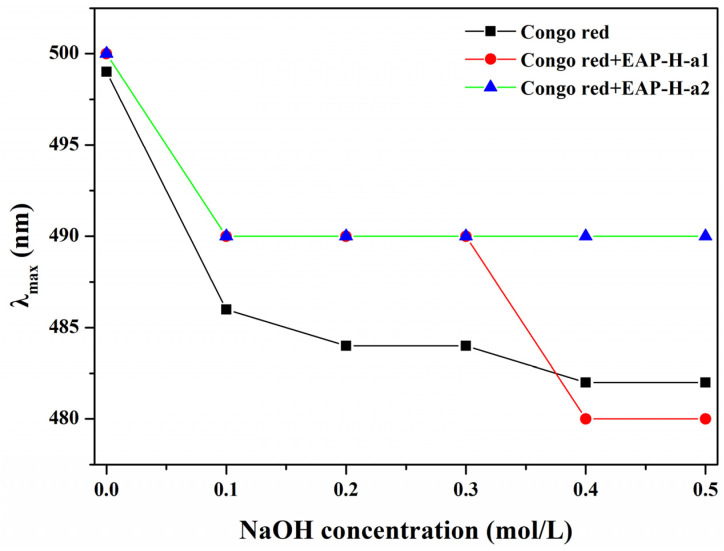
Congo red analysis of the polysaccharides.

**Figure 5 molecules-27-06415-f005:**
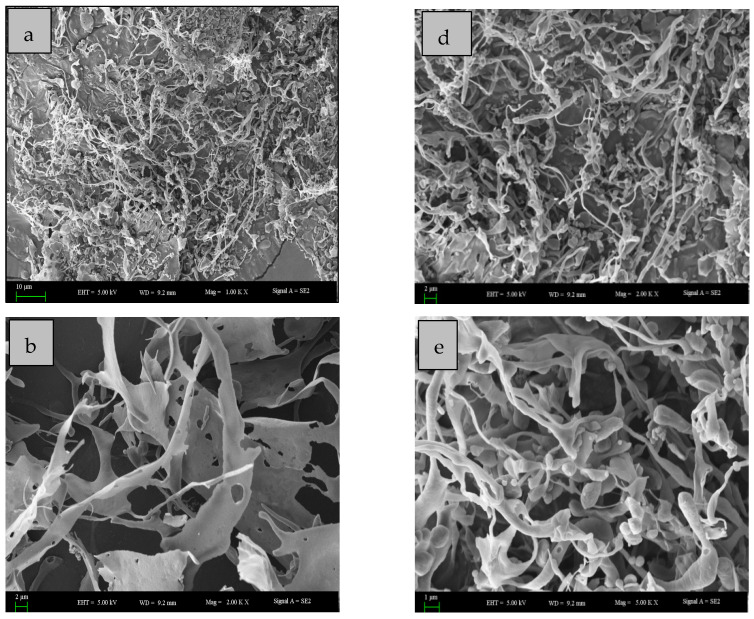
Scanning electron microscopy picture EAP-H-a2 (**a**–**c**); EAP-H-a1 (**d**–**f**).

**Figure 6 molecules-27-06415-f006:**
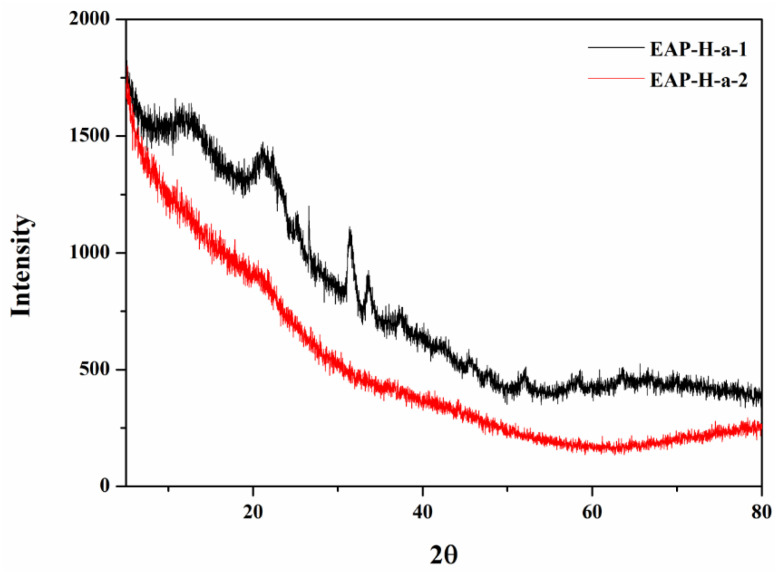
XRD spectrum of the EAP-H-a1 and EAP-H-a2.

**Figure 7 molecules-27-06415-f007:**
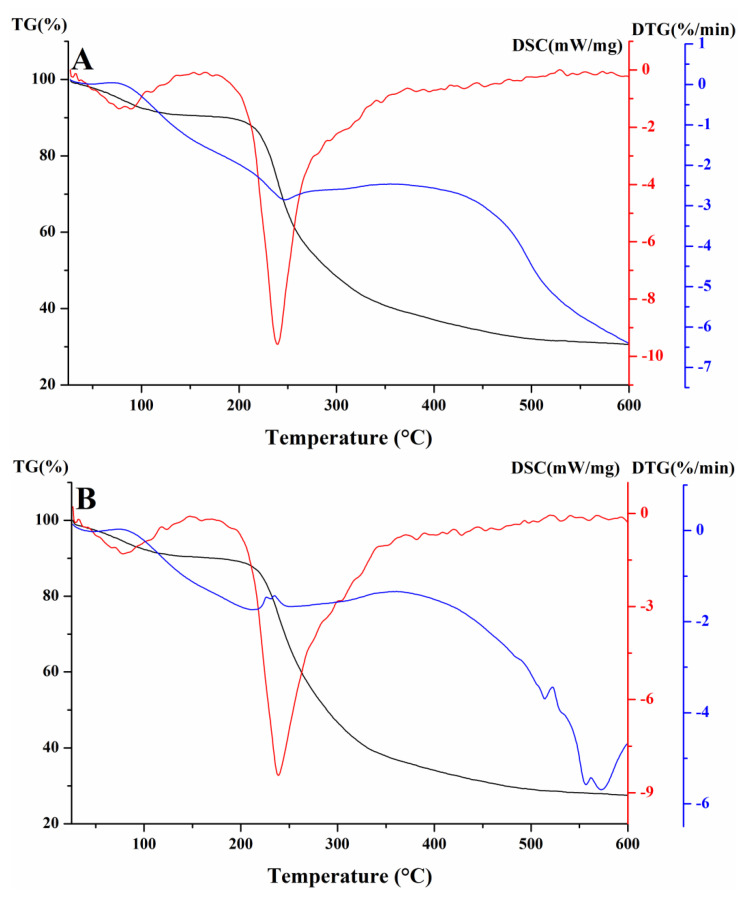
Thermal properties of EAP-H-a1 (**A**) and EAP-H-a2 (**B**).

**Figure 8 molecules-27-06415-f008:**
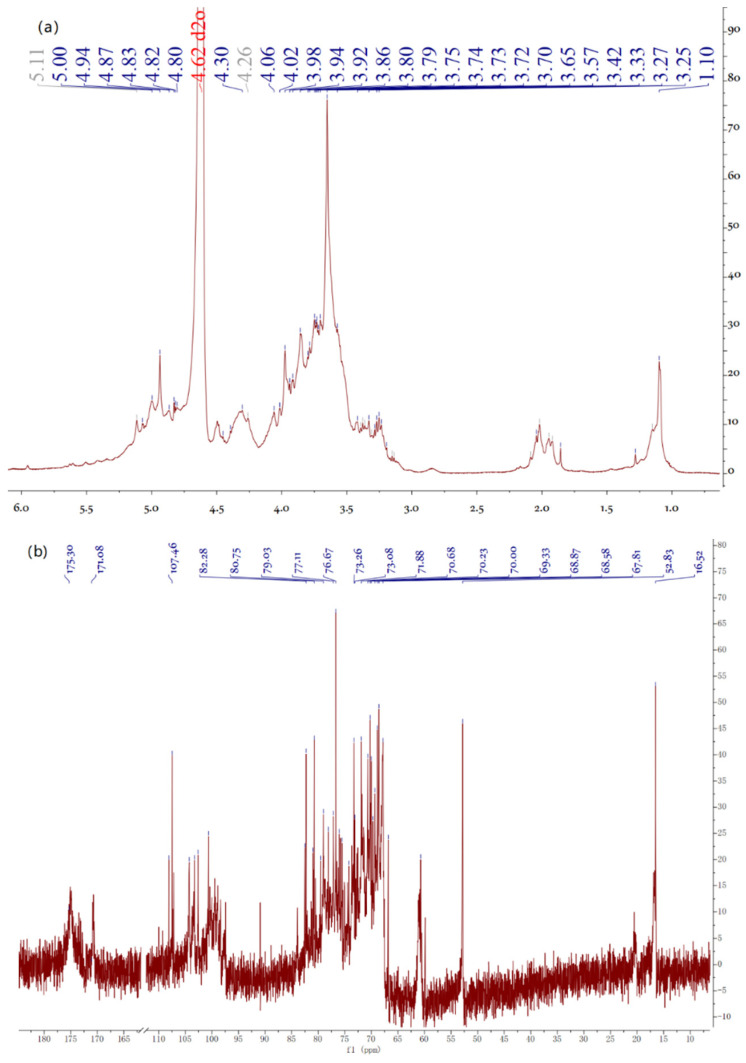
^1^H (**a**)and ^13^C (**b**) spectra of EAP-H-a2.

**Figure 9 molecules-27-06415-f009:**
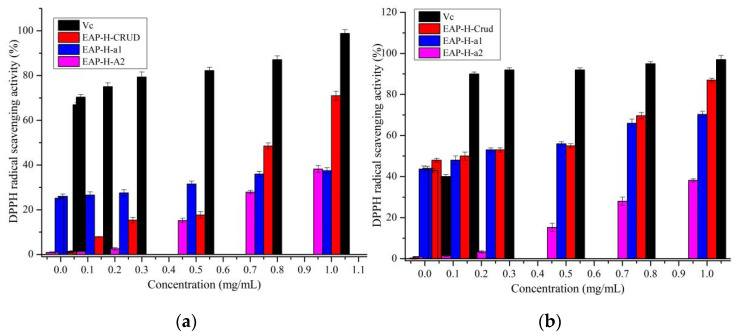
Antioxidant activity of the polysaccharides. (**a**) DPPH; (**b**) ABTS.

**Figure 10 molecules-27-06415-f010:**
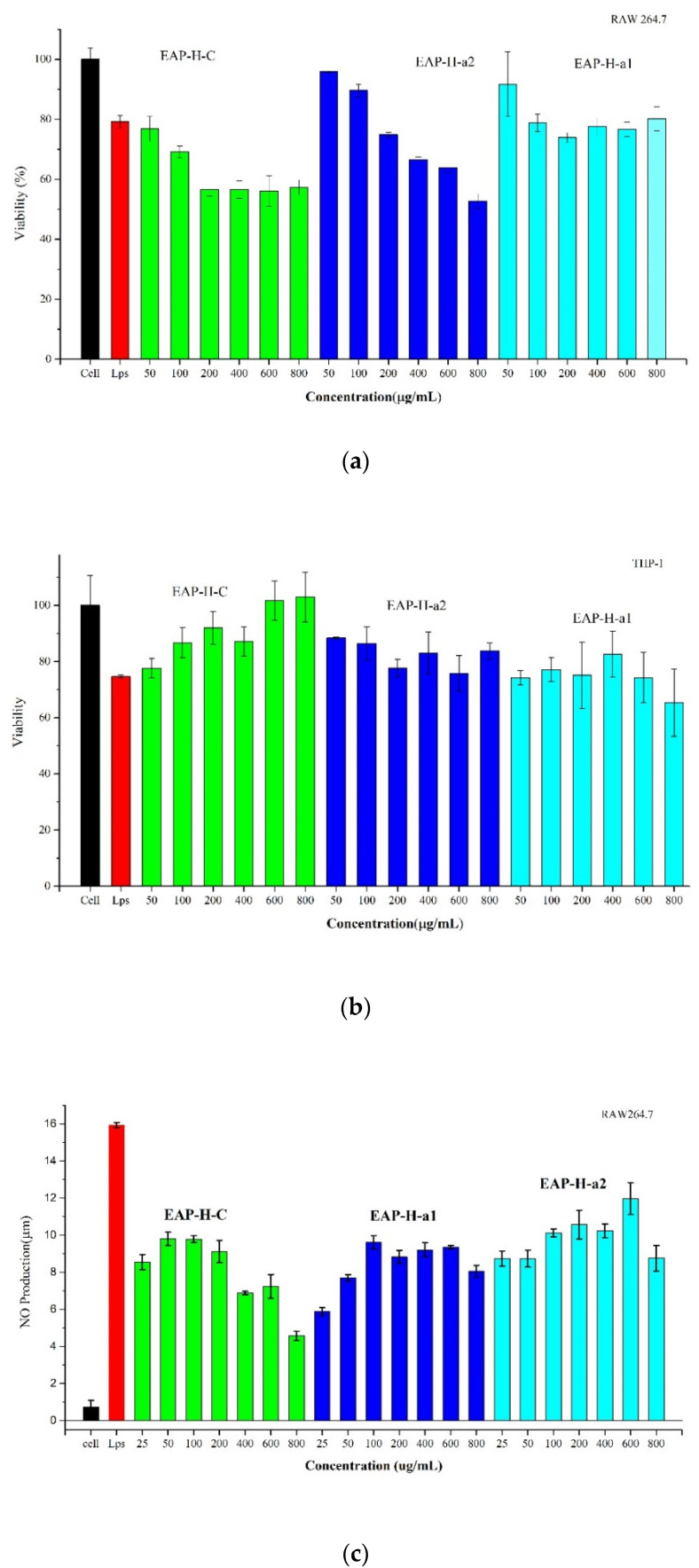
Effects of different concentrations of polysaccharides on the cell viabilities on RAW 264.7 (**a**) and THP-1 (**b**); NO production on RAW 264.7 (**c**); phagocytic capacity on RAW 264.7 (**d**); and THP-1 (**e**).

**Table 1 molecules-27-06415-t001:** Monosaccharide composition and contents of the purified fractions.

	EAP-H-a1	EAP-H-a2
Neutral sugar (%)	12.55	12.24
Protein (%)	0	0
Uronic acid (%)	67.15	62.24
Mw (kDa)	705.79	439.85
[α]20 D (c 1.0 mg/mL, H_2_O)	+290.2	+175.9
**Monosaccharide Composition (Molar Ratio)**		
Rha	13.752	24.861
Ara	20.574	19.746
Xyl	23.361	8.282
Glc	8.894	8.474
Gal	33.418	38.637

Abbreviations: Rha, rhamnose; Ara, arabinose; Xyl, xylose; Man, mannose; Glc, glucose; Gal, galactose.

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
