# Peer review of "Isolation, Structural Characterization, and Biological Activity of the Two Acidic Polysaccharides from the Fruits of the *Elaeagnus angustifolia* Linnaeus"

_molecules, 2022, doi:10.3390/molecules27196415_

Round 1
Reviewer 1 Report
The work entitled "Isolation, Structural Characterization, and Biological Activity of the Two Acidic Polysaccharides from the Fruits of the Elaeagnus Angustifolia L." by Huojiaaihemaiti et al describes an original research on the extraction of polysaccharides from the fruits of E. angustifolia. The study is well designed and it employed various analytical techniques such as GC-MS, HPLC, XRD and SEM for the structural elucidation of the polysaccharides. Moreover, the antioxidant, immuno-regulatory and anti-inflammatory properties of the isolated polysaccharides were investigated. The study is very relevant to the readership of Molecules and is properly conducted. The study can be considered for publication after considering the following points. One major flaw is the lack of any evidence of statistical analysis on the presented bioassay data as histograms. I list the following points that need to be addressed:
-line 30: replace key word "structure" by other relevant keywords such as "structural elucidation" or "Ion-exchange chromatography".
-line 49: immune regulating change to immune-regulating, with hyphen between.
- line 51: replace "research" with "reports".
-line 54; replace "the" by "this medicinal plant".
-Paragraph (lines 47-50): authors need to elaborate on the previously reported biological activities such as anticancer and antibacterial to give the reader a background knowledge of what has been previously known so far about the biological activities of the polysaccharides separated from the fruits of E. angustifolia L.
-Paragraph (lines 51-53): authors stated that "To our best knowledge, there are few research about the systematic isolation, structural characterization, and biological activities of acidic polysaccharide composition from the fruits of E. angustifolia L.". Proper citation of the previous reports and what specific activities have been conducted. Also, authors need to point the novel aspect of the experimental part conducted here relative to what has been previously reported.
-Were the immuno-regulatory and anti-inflammatory activities of the isolated polysaccharides previously reported??.
-Would the authors provide statistical analysis with p values on all the described histograms. This is very important to show whether the observed differences in treatments are significantly different from controls. Therefore, proper statistical analysis is required.
-Why the authors did not calculate the IC50 values for the MTT assay??. This is very important as a reference value to reflect how cytotoxic is a given constituent.
Author Response
Response to Reviewers
Dear Editors and Reviewers:
Thank you for your letter and for the reviewers’ comments concerning our manuscript entitled “Isolation, Structural Characterization, and Biological Activity of the Two Acidic Polysaccharides from the Fruits of the Elaeagnus Angustifolia L.” (ID: molecules-1915824). Those comments are all valuable and very helpful for revising and improving our paper, as well as the important guiding significance to our researches. We have studied comments carefully and have made correction which we hope meet with approval. Revised portion are marked in red in the paper. The main corrections in the paper and the responds to the reviewer’s comments are as flowing:
Responds to the reviewer’s comments:
-line 30: replace key word "structure" by other relevant keywords such as "structural elucidation" or "Ion-exchange chromatography".
Thanks for the reviewer’s comment, "structure" was replaced by "structural elucidation" at the keywords in revised manuscript.
-line 49: immune regulating change to immune-regulating, with hyphen between.
Thanks for the reviewer’s comment, "immune regulating" was change to "immune regulating" with hyphen in revised manuscript.
- line 51: replace "research" with "reports".
Thanks for the reviewer’s comment, "research" was change to "reports" in revised manuscript.
-line 54; replace "the" by "this medicinal plant".
Thanks for the reviewer’s comment, it was corrected in revised manuscript.
-Paragraph (lines 47-50): authors need to elaborate on the previously reported biological activities such as anticancer and antibacterial to give the reader a background knowledge of what has been previously known so far about the biological activities of the polysaccharides separated from the fruits of E. angustifolia L.
Thanks for the reviewer’s comment, it was added in revised manuscript.
-Paragraph (lines 51-53): authors stated that "To our best knowledge, there are few research about the systematic isolation, structural characterization, and biological activities of acidic polysaccharide composition from the fruits of E. angustifolia L.". Proper citation of the previous reports and what specific activities have been conducted. Also, authors need to point the novel aspect of the experimental part conducted here relative to what has been previously reported.
-Were the immuno-regulatory and anti-inflammatory activities of the isolated polysaccharides previously reported??.
Thanks for the reviewers comment. The previous reports about the polysaccharides from the fruits of the E. angustifolia L. was mainly focused on the extraction technology optimization, structural characterization and biological activity .analysis such as antioxidant, antibacterial, antitumor, and immune-regulating functions of neutral polysaccharides. The structural characterization was conducted by the analysis of neutral polysaccharides with UV, FT-IR, HPLC, GC-MS, SEM and XRD et al. In this research we have elucidated the structure of the obtained two acidic polysaccharide through the combination of UV, FT-IR, GC-MS, HPGPC, XRD, DSC, TG, DTG, SEM, congo red as well as NMR analysis.
There have one report about the immuno-regulatory activities of the isolated neutral polysaccharides which previously reported. However, there found different structure between the reported one with ours. There are different activities.
-Would the authors provide statistical analysis with p values on all the described histograms. This is very important to show whether the observed differences in treatments are significantly different from controls. Therefore, proper statistical analysis is required.
Thanks for the reviewers comment. We are sorry to make this mistake. We have corrected it in the revised manuscript.
-Why the authors did not calculate the IC50 values for the MTT assay??. This is very important as a reference value to reflect how cytotoxic is a given constituent.
Thanks for the reviewers comment. In this paper, we have screened the immuno-regulating activity of the isolated polysaccharide fractions through MTT assay. aim of MTT assay in our paper is to evaluate polysaccharides effect on cell growth. Our polysaccharide at high concentration can not reach IC50 . The depth research about mechanisim of immu-nomodulatory and anti-inflammatory activity will be studied in our further experiments.

Reviewer 2 Report
I am honored to participate in the valuation of the article entitled : Isolation, Structural Characterization, and Biological Activity of the Two Acidic Polysaccharides from the Fruits of the Elaeagnus Angustifolia L.
This work has been well presented and the subject and purpose has been important to publish from this magazine of Moleules. But it is necessary to make corrections before publishing among the following comments:
Abstract
It is necessary to add information about the use in traditional phytotherapy.
It is necessary to define the abbreviations
Generally, this section is considered important in the article, so it is necessary to improve this section respecting the order of presentation, materials, results and conclusion.
Keywords : use words that are not mentioned in the title
Matriels et methode
‘’ 3.1. Materials and reagents’’ : it is better to separate the plant material and the chemical products used. Please add the name of the botanist who made the identification
Please rephrase this sentence: ‘’ Dried E. Angustifolia L. fruit was soaked in 95% ethanol at room temperature for 24 h to remove small molecules and fruit seed and air-dried to give the desired powder. The dried fruit of E. Angustifolia L (1000 g) was extracted with sequential extraction with water (room temperature), hot water, acid, and alkaline, respectively.’’
‘’3.3. Purification of the EAP-H : Quantity of NaCl’’
‘’3.5. Antioxidant activity’’ : Please detail this part, the dilution, the solubility...
Results and Discussion
- please, use the same fonts for the Figures
- check the citation of the Figures
Author Response
Response to Reviewers
Dear Editors and Reviewers:
Thank you for your letter and for the reviewers’ comments concerning our manuscript entitled “Isolation, Structural Characterization, and Biological Activity of the Two Acidic Polysaccharides from the Fruits of the Elaeagnus Angustifolia L.” (ID: molecules-1915824). Those comments are all valuable and very helpful for revising and improving our paper, as well as the important guiding significance to our researches. We have studied comments carefully and have made correction which we hope meet with approval. Revised portion are marked in red in the paper. The main corrections in the paper and the responds to the reviewer’s comments are as flowing:
Responds to the reviewer’s comments:
I am honored to participate in the valuation of the article entitled : Isolation, Structural Characterization, and Biological Activity of the Two Acidic Polysaccharides from the Fruits of the Elaeagnus Angustifolia L.
This work has been well presented and the subject and purpose has been important to publish from this magazine of Moleules. But it is necessary to make corrections before publishing among the following comments:
Abstract
It is necessary to add information about the use in traditional phytotherapy.
Thanks for the reviewer’s comment, it was added in abstract of revised manuscript.
It is necessary to define the abbreviations
Generally, this section is considered important in the article, so it is necessary to improve this section respecting the order of presentation, materials, results and conclusion.
Keywords : use words that are not mentioned in the title
Thanks for the reviewer’s comment, it was added in revised manuscript.
Materials et methods
‘’ 3.1. Materials and reagents’’ : it is better to separate the plant material and the chemical products used. Please add the name of the botanist who made the identification.
Thanks for the reviewer’s comment, “Materials and reagents” section was rewritten and the botanist made the identification was also added in the revised manuscript.
Please rephrase this sentence: ‘’ Dried E. Angustifolia L. fruit was soaked in 95% ethanol at room temperature for 24 h to remove small molecules and fruit seed and air-dried to give the desired powder. desired powder of E. Angustifolia L (1000 g) fruit was extracted with sequential extraction with water (room temperature), hot water, acid, and alkaline, respectively.’’
Thanks for the reviewer’s comment, This sentence was corrected in the manuscript as follows:
The small molecules of the dried fruits of the E. Angustifolia L. was removed through soaking in 95% ethanol at room temperature for 24 h. The obtained powder (1000 g) was extracted by sequential extraction with water (room temperature 25°C), hot water (85°C), acid, and alkaline, respectively.
‘’3.3. Purification of the EAP-H : Quantity of NaCl’’
Thanks for the reviewer’s comment, EAP-H was purified with different concentration of NaCl, it was firstly purified with 0.2 M NaCl and monitored with the anthrone-sulfuric method, when the fraction was thoroughly eluted, we have changed to 0.4 M Na Cl as the eluent, and completed the purification in this way.
‘’3.5. Antioxidant activity’’ : Please detail this part, the dilution, the solubility...
Thanks for the reviewer’s comment, we have corrected it in the revised manuscript.
Results and Discussion
- please, use the same fonts for the Figures
Thanks for the reviewer’s comment, it was corrected in revised manuscript.
- check the citation of the Figures
Thanks for the reviewer’s comment, it was checked and revised in the submitted manuscript.
Reviewer 3 Report
In the present paper authors describe isolation, structural characterization, and biological activity of the two polysaccharides from the plant Elaeagnus Angustifolia L.
Technically there are many problems all the way through the text (example: the legend in Figure 9a is wrong, under Figure 10 there is no description, variables throughout the text are not in italics…).
Introduction needs to be improved, it is too short and does not provide proper introduction to the topic.
Results and discussion section describes results but does not discuss obtained results in detail. This is not acceptable, so it must be improved or rewritten.
Materials and methods section is too short and not properly described.
Conclusions are not supported by the obtained results.
Overall merit is very low, paper is badly written, it lacks novelty and important experimental information, so I do not recommend this paper to be published in this journal.
Author Response
Response to Reviewers
Dear Editors and Reviewers:
Thank you for your letter and for the reviewers’ comments concerning our manuscript entitled “Isolation, Structural Characterization, and Biological Activity of the Two Acidic Polysaccharides from the Fruits of the Elaeagnus Angustifolia L.” (ID: molecules-1915824). Those comments are all valuable and very helpful for revising and improving our paper, as well as the important guiding significance to our researches. We have studied comments carefully and have made correction which we hope meet with approval. Revised portion are marked in red in the paper. The main corrections in the paper and the responds to the reviewer’s comments are as flowing:
Responds to the reviewer’s comments:
In the present paper authors describe isolation, structural characterization, and biological activity of the two polysaccharides from the plant Elaeagnus Angustifolia L.
Technically there are many problems all the way through the text (example: the legend in Figure 9a is wrong, under Figure 10 there is no description, variables throughout the text are not in italics…).
Thanks for the reviewer’s comment. We are sorry to make this mistake. We have corrected it in the revised manuscript.
Introduction needs to be improved, it is too short and does not provide proper introduction to the topic.
Thanks for the reviewer’s comment. We have made some corrections in the revised manuscript.
Results and discussion section describes results but does not discuss obtained results in detail. This is not acceptable, so it must be improved or rewritten.
Thanks for the reviewer’s comment. We are very sorry for that. Reference material about our two novel acidic polysaccharides is very limited which give us difficulties to discuss obtained results in detail. We have made efforts to correct it in the revised manuscript. may be this not enough to satisfy reviewer’s comment. We are sorry. This is a part of our research related to systematic analysis about polysaccharides of Elaeagnus Angustifolia L The depth research about them (methylation analysis, monosaccaride limkage analysis,polysaccaride extraction with acid and base with their structral and biological analysis ) will be studied in our further experiments.
Materials and methods section is too short and not properly described.
Thanks for the reviewer’s comment. We are sorry to make this mistake. We have corrected it in the revised manuscript.
Conclusions are not supported by the obtained results.
Thanks for the reviewer’s comment. We are sorry to make this mistake. We have corrected it in the revised manuscript. The conclusion was re-written as follows:
In this study, sequential extractions of polysaccharides from E. angustifolia L. were performed and the hot water extracted one was further purified by anion exchange and gel permeation chromatography. The valuable structural information about the physicochemical properties of the isolated two acidic polysaccharides were obtained by using different analytical tools and methods. EAP-H-a1 and EAP-H-a2 were homogenous acidic polysaccharides with an average Mw of 705.796 kDa and 439.852 kDa respectively, which composed of Rha, Xyl, Ara, Man, Glc, and Gal in a different molar ratios. Both have certain free radical scavenging activity against DPPH and ABTS, demonstrating their anti-oxidant function. The obtained acidic polysaccharides could promote the secretion of NO and phagocytic activities of RAW 264.7 and THP-1. The overall study suggested that acidic EAP-H could be used as a potential natural antioxidant and immuno-regulating drug resource and has value for further scientific research in the food and pharmaceutical industry.
Overall merit is very low, paper is badly written, it lacks novelty and important experimental information, so I do not recommend this paper to be published in this journal.

Reviewer 4 Report
The manuscript reports a good level of data.
The abbreviations should be deciphered at first mention, for example:
line 21- Fourier-transform infrared spectroscopy (FTIR)
line 23 - SEM .... and XRD …. analysis
The minor check is required for the English language and style:
lines 34-35 (the word "species" occurs three times in one sentence): Elageanus species belong to Elaeagnaceae Family that has 55 species in China [1] and 34 three species in Xinjiang Autonomous Region China.
lines 53-61 – The purpose of the research should be rephrased, because the prospect of studying antioxidant activity, which will be presented later in the results, is not mentioned.
Line 58 - Meanwhile, immuno-regulatory and anti-inflammatory activities of the obtained polysaccharides (should be rephrased).
line 36 - Elaeagnus Oxycarpa (word 'oxycarpa' must start with a lowercase letter) as well as 'Angustifolia L' in the title of paper.
lines 41, 53, 60 etc. - The surname of the scientist who described the species E. angustifolia for science (L. - Linnaeus) should be indicated after the name of the species only at the first mention (line 35), and it does not need to be used further on in the text.
The Introduction section should be expanded and supplemented with the new references to demonstrate the relevance of such studies. I assume that this sentence needs confirmation in the form of literary references (lines52-53) 'To our best knowledge, there are few research about the systematic isolation, structural characterization, and biological activities of acidic polysaccharide composition from the fruits of E. angustifolia L. [...]
Several articles on this topic 'Elaeagnus angustifolia Polysaccharides' could be found in the PubMed database. The authors can use them to expand the introduction:
https://pubmed.ncbi.nlm.nih.gov/?term=Elaeagnus+angustifolia+Polysaccharides
Іt would be good to paraphrase the Conclusions a little for a clear understanding
The italic type should be used everywhere for writing Latin names of genus and species:
line 402 - (Elaeagnus angustifolia L.), etc.
Author Response
Response to Reviewers
Dear Editors and Reviewers:
Thank you for your letter and for the reviewers’ comments concerning our manuscript entitled “Isolation, Structural Characterization, and Biological Activity of the Two Acidic Polysaccharides from the Fruits of the Elaeagnus Angustifolia L.” (ID: molecules-1915824). Those comments are all valuable and very helpful for revising and improving our paper, as well as the important guiding significance to our researches. We have studied comments carefully and have made correction which we hope meet with approval. Revised portion are marked in red in the paper. The main corrections in the paper and the responds to the reviewer’s comments are as flowing:
Responds to the reviewer’s comments:
The manuscript reports a good level of data.
The abbreviations should be deciphered at first mention, for example:
line 21- Fourier-transform infrared spectroscopy (FTIR)
line 23 - SEM .... and XRD …. analysis
Thanks for the reviewer’s comment, it was corrected in revised manuscript.
The minor check is required for the English language and style:
Thanks for the reviewer’s comment. We are sorry to make this mistake. We have corrected it in the revised manuscript.
lines 34-35 (the word "species" occurs three times in one sentence): Elageanus species belong to Elaeagnaceae Family that has 55 species in China [1] and 34 three species in Xinjiang Autonomous Region China.
Thanks for the reviewer’s comment. We are sorry to make this mistake. We have corrected it in the revised manuscript.
lines 53-61 – The purpose of the research should be rephrased, because the prospect of studying antioxidant activity, which will be presented later in the results, is not mentioned.
Thanks for the reviewer’s comment, it was corrected in revised manuscript as follows:
Therefore, the purpose of the present study was to isolation, structural elucidation, and evaluate the biological activity of this medicinal plant.
Line 58 - Meanwhile, immuno-regulatory activities of the obtained polysaccharides (should be rephrased).
Thanks for the reviewer’s comment, it was corrected in revised manuscript.
line 36 - Elaeagnus Oxycarpa (word 'oxycarpa' must start with a lowercase letter) as well as 'Angustifolia L' in the title of paper.
Thanks for the reviewer’s comment. We are sorry to make this mistake. We have corrected it in the revised manuscript.
lines 41, 53, 60 etc. - The surname of the scientist who described the species E. angustifolia- Linnaeus for science (L. - Linnaeus) should be indicated after the name of the species only at the first mention (line 35), and it does not need to be used further on in the text.
Thanks for the reviewer’s comment. We are sorry to make this mistake. We have corrected it in the revised manuscript.
The Introduction section should be expanded and supplemented with the new references to demonstrate the relevance of such studies. I assume that this sentence needs confirmation in the form of literary references (lines52-53) 'To our best knowledge, there are few research about the systematic isolation, structural characterization, and biological activities of acidic polysaccharide composition from the fruits of E. angustifolia L.
Thanks for the reviewer’s comment. It was corrected in revised manuscript.
Several articles on this topic 'Elaeagnus angustifolia Polysaccharides' could be found in the PubMed database. The authors can use them to expand the introduction:
Thanks for the reviewer’s comment.
Іt would be good to paraphrase the Conclusions a little for a clear understanding
Thanks for the reviewer’s comment, it was corrected in revised manuscript as follows:
In this study, sequential extractions of polysaccharides from E. angustifolia L. were performed and the hot water extracted one was further purified by anion exchange and gel permeation chromatography. The valuable structural information about the physicochemical properties of the isolated two acidic polysaccharides were obtained by using different analytical tools and methods. EAP-H-a1 and EAP-H-a2 were homogenous acidic polysaccharides with an average Mw of 705.796 kDa and 439.852 kDa respectively, which composed of Rha, Xyl, Ara, Man, Glc, and Gal in a different molar ratios. Both have certain free radical scavenging activity against DPPH and ABTS, demonstrating their anti-oxidant function. The obtained acidic polysaccharides could promote the secretion of NO and phagocytic activities of RAW 264.7 and THP-1. The overall study suggested that acidic EAP-H could be used as a potential natural antioxidant and immuno-regulating drug resource and has value for further scientific research in the food and pharmaceutical industry.
The italic type should be used everywhere for writing Latin names of genus and species:
line 402 - (Elaeagnus angustifolia L.), etc.
Thanks for the reviewer’s comment. We are sorry to make this mistake. We have corrected it in the revised manuscript.
Round 2
Reviewer 1 Report
I have no further comments
Reviewer 3 Report
This paper still has many technical errors that have not yet been corrected
- variables throughout the text are not in italics
- there is no space between the variable and the measuring unit in some cases
- in the description of Figure 2 GC is mentioned, and in the text the HPLC spectrum is described. Please correct this.
Introduction has been improved, so now it provides proper introduction to the topic.
The results and discussion chapter still has shortcomings, but I accept arguments from the authors why it cannot be written more extensively at the moment.
Materials and methods section as well as conclusions have been improved.
With the corrections made, this paper is significantly improved, but before publication all technical errors along the text must be corrected. I ask the authors to go through the entire text in detail and correct all technical errors.